# The Effects of Filgrastim and Hyaluronic Acid on the Endometrium in Experimentally Induced Asherman Syndrome Rat Models: A Prospective Laboratory Study [note 1]

**DOI:** 10.3390/jcm14103334

**Published:** 2025-05-11

**Authors:** Merve Genco, Mehmet Genco, Fisun Vural, Nermin Koç

**Affiliations:** 1Department of Obstetrics and Gynecology, Haydarpaşa Numune Training and Research Hospital, Istanbul 34668, Turkey; m.genco@hotmail.com (M.G.); fisunvural@yahoo.com.tr (F.V.); 2Department of Pathology, Haydarpaşa Numune Training and Research Hospital, Istanbul 34668, Turkey; nerminkoc@yahoo.com

**Keywords:** filgrastim, hyaluronic acid, endometrium, Asherman syndrome, VEGF, granulocyte-colony stimulating factor

## Abstract

**Background/Objectives:** The aim of the present study was to explore the histopathological effects and tissue Vascular Endothelial Growth Factor (VEGF) levels of filgrastim and hyaluronic acid treatment in a rat model with experimentally induced Asherman syndrome. **Methods:** In this study, 26 female Sprague Dawley rats were used. First, a rat model of Asherman syndrome was established in two rats, and the remaining rats were randomly divided into three groups. A total of 0.1 mL trichloroacetic acid was applied to the right uterine horns of all groups to induce adhesion formation.Group I received no treatment, Group II received intrauterine hyaluronic acid treatment (0.01), Group III received subcutaneous Filgrastim treatment (50 μg/kg/day), and Group IV received both intrauterine hyaluronic acid and subcutaneous Filgrastim treatment. Histopathological analysis of uterine horns in the rats with and without Asherman syndrome, inflammation, glandular count, and fibrosis levels were examined. Tissue VEGF levels were investigated immunohistochemically. **Results:** Hyaluronic acid treatment resulted in an increase only in uterine lumen diameter and VEGF levels, while Filgrastim treatment led to an increase in uterine wall diameter, lumen diameter, gland count, and VEGF levels, as well as a decrease in fibrosis and inflammation scores. Combined treatment with filgrastim and hyaluronic acid showed an increase in lumen diameter, gland count, and VEGF levels, along with a decrease in inflammation and fibrosis scores (*p* < 0.05). Filgrastim treatment resulted in better effects for Asherman syndrome compared to hyaluronic acid treatment. There were no beneficial effects seen with the combined therapy. **Conclusions:** Filgrastim treatment resulted in better outcomes for Asherman syndrome compared to hyaluronic acid treatment. The combined therapy did not show additional benefits beyond what was achieved with Filgrastim treatment alone.

## 1. Introduction

Adhesions or scar tissue formation within the uterine cavity is called intrauterine adhesions (IUAs) or Asherman syndrome. Heinrich Fritsch was the first to define IUAs and related symptoms in 1894. Later, Josef Asherman introduced the definition of Asherman syndrome in 1948 [1,2]. Since then, several cases and treatment protocols have been proposed. Patients present with a wide variety of symptoms such as infertility, hypomenorrhea, amenorrhea, menstrual irregularities, pregnancy loss, obstetric complications, and placental invasion anomalies [3]. The diagnostic criteria for Asherman syndrome diagnosis and therapeutic approaches are still contested.

The incidence of Asherman syndrome varies based on the population studied, with an incidence of 4.6% in infertile women, 37.6% in women who have an abortion, and 40% in women who have a recurrent curettage. Trauma and infections of the endometrium basal layer are the leading causes of IUA. The pathogenesis of Asherman syndrome is complex and is not limited to tissue damage, infection, or inflammation. Immune system cells also play an active role in this process. Specifically, the activation of immune system cells during curettage and tissue damage can disrupt endometrial healing, contributing to adhesion formation. Curettage, myomectomy, Mullerian anomaly surgeries, uterine embolization and B-lynch sutures, genital tuberculosis, and schistosomiasis are known risk factors [4,5].

Imaging methods such as transvaginal and transabdominal ultrasonography, sonohysterography, or hysterosalpingography provide insights into adhesions [6,7]. Hysteroscopy is the gold standard for diagnosis and treatment [8]. However, re-adhesion formation after surgery is a significant problem. Minimally invasive treatment approaches that prevent the formation of adhesion have been attempted, such as the use of intrauterine devices, Foley catheters, and gel barriers [5,8,9,10,11,12]. The success rate of adhesiolysis in patients with menstrual problems is 52–88%, and live birth rates in infertile patients vary between 25–35%. The ratio of re-adhesion development ranges from 52–62% in cases with severe adhesions. Thus, additional treatment approaches are required following surgery.

Trichloroacetic acid is a chemical agent used to induce endometrial damage in models of Asherman syndrome. When applied intrauterinely, trichloroacetic acid creates a controlled chemical burn on the endometrial surface, leading to tissue necrosis. This necrotic process disrupts the normal regenerative capacity of the endometrium, resulting in the formation of adhesions. These adhesions mimic the obliteration of the uterine cavity, which is the pathological hallmark of Asherman syndrome. Therefore, trichloroacetic acid is commonly used in research to establish a model of Asherman syndrome reliably [9].

Granulocyte colony-stimulating factor (G-CSF) is a hematopoietic-specific cytokine produced by hematopoietic cells, stromal cells, fibroblasts, and macrophages [13]. However, it is also present in a few non-hematopoietic cell types, such as endothelial cells, placental cells, trophoblasts, and granulosa-lutein cells [14,15]. G-CSF primarily promotes the migration and differentiation of stem cells, facilitating endometrial regeneration by promoting angiogenesis and reducing apoptotic activity [16]. G-CSF transiently suppresses the immune response through its effects on lymphocytes, macrophages, and T helper-2 cells [17]. G-CSF has been used in studies on human thin endometrium [18,19]. In recent years, successful results in studies related to recurrent pregnancy loss and repeated implantation failure have been reported [20,21]. Studies have shown that G-CSF increases implantation and pregnancy rates, although there are also studies that show that G-CSF application does not change endometrial and pregnancy rates. Studies have also shown that G-CSF increases endometrial thickness and shapes the endometrium [18,19,22,23,24]. Filgrastim is a form of G-CSF and is used for the mobilization of hematopoietic stem cells.

Hyaluronic acid is a biologically stable, non-toxic, high-molecular-weight polyanionic polysaccharide. Hyaluronic acid has been shown to improve wound healing without causing excessive connective tissue formation. In a recent study, intraperitoneal administration of hyaluronic acid significantly reduced adhesions [25]. Hyaluronic acid gel has been widely used to prevent adhesion formation after surgical adhesiolysis, but its effect on pregnancy rates is a subject of controversy.

Asherman syndrome is defined as the formation of intrauterine adhesions resulting from endometrial injury, which may partially or completely obstruct the uterine cavity. During this process, endometrial blood flow and regenerative capacity are compromised, hindering the reconstruction and functional recovery of the endometrium. Vascular Endothelial Growth Factor (VEGF) plays a key role in promoting angiogenesis and tissue repair, particularly in endometrial tissue regeneration. Increased VEGF levels can stimulate angiogenesis in damaged tissue, supporting improved blood flow and tissue repair. In Asherman syndrome, optimizing endometrial healing is critical, and VEGF may be essential in this process. Therefore, assessing VEGF expression can serve as a valuable biomarker for evaluating treatment response and understanding endometrial regeneration [9].

Experimental studies on Asherman syndrome are limited. In particular, the potential effect of Filgrastim and hyaluronic acid treatment combined has not been assessed. Thus, in the present study, the effect of Filgrastim alone or combined with hyaluronic acid therapy was assessed in a rat model of Asherman syndrome.

Asherman syndrome is primarily characterized by intrauterine adhesions caused by endometrial damage, which disrupts the regenerative process of the endometrium. This damage impairs blood flow, angiogenesis, and tissue repair mechanisms. Effective therapeutic interventions are essential for preventing adhesion formation and promoting tissue healing. G-CSF and hyaluronic acid have shown promise as treatment modalities in this context due to their distinct but complementary roles in tissue regeneration. While G-CSF enhances angiogenesis, stem cell migration, and immune modulation, hyaluronic acid reduces scar formation and promotes wound healing. The combination of these treatments may potentially address the complex pathology of Asherman syndrome more effectively.

## 2. Materials and Methods

The present study was approved by the Yeditepe University Animal Ethics Committee (approval no. HADYEK: 2022/01-2, approved on 28 January 2022). All procedures involving animals were performed in accordance with the ethical guidelines and humane practices outlined by the Yeditepe University Animal Ethics Committee. Every effort was made to minimize animal suffering, and all experimental protocols were performed in compliance with internationally accepted standards for the care and use of laboratory animals. The study involved 26 Sprague-Dawley female rats weighing between 200 and 250 g, which were not pregnant. The animals were acclimated in the research center for 1 week before initiation of the study, housed in a suitable environment with proper ventilation, and subjected to a 12 h light/dark cycle while being fed according to standards. Vaginal cytology was used to assess progression through the estrous cycle.

The rat model was selected for inducing Asherman syndrome due to rats’ short reproductive cycles, well-defined uterine anatomy, adaptability to experimental conditions, and cost-effectiveness, making them commonly used animals for studies involving experimental induction of uterine pathologies. Additionally, the rat uterus shares histological and physiological similarities with the human uterus, facilitating the translation of experimental results into clinical practice.

The experimental model of Asherman syndrome was established using a surgical procedure established by Jing et al. [26] and Hunter et al. [27] during laparotomy. Initially, the model was established in two rats to ensure the procedure worked. Xylazine hydrochloride (6 mg/kg; Xylazine-HCl, Rompun, Bayer AG, Leverkusen, Germany) and Ketamine hydrochloride (85 mg/kg; Keta-control, Mefar Ilac^®^, Pendik, Turkey) were administered intramuscularly for anesthesia. After shaving and properly cleaning the area, laparotomy was performed using a 2 cm incision. In the rat’s uterus with a bicornuate appearance, each horn was opened into the vagina with a separate cervical canal. The right uterine horn was held at the top and at the cervical level with non-crushing clamps. A total of 0.1 mL trichloroacetic acid (IL 33^®^, Istanbul Ilac Sanayi ve Ticaret AS, Umraniye, Turkey) was injected into the right uterine horn with an insulin syringe. Trichloroacetic acid was used as described previously by Kılıç et al. [28]. After the application of trichloroacetic acid, a sponge was used to prevent the acid from leaking around, and the abdomen was washed with saline. No procedure was performed on the left uterine horn. The muscle layer and skin were closed. The rats were then maintained for 14 days as described above. Then, the rats were sacrificed, and their uteri were used for histopathological examination (Figure 1). A treatment experiment was planned with the 26 rats, but five rats died prior to laparotomy. The deaths observed in the experimental groups were attributed to the systemic effects of trichloroacetic acid, which resulted in a chemical injury to establish the Asherman syndrome model. No signs of infection were detected in these animals during postmortem examinations. For this reason, in the present study, there were seven rats in Group 2 and six rats in the other groups.

*Treatment groups.* After trichloroacetic acid application, treatments were applied 2 weeks later (three estrous cycles): For Group 1, no additional treatment was given. For *Group 2*, *a* 1”second laparotomy was performed 14 days (three estrous cycles) after inducing Asherman syndrome. A total of 0.01 mL low-molecular-weight hyaluronic acid was applied to the right uterine horn, close to the ovary, towards the cervix. No procedure was performed on the left uterine horn. For *Group 3*, 14 days after the establishment of Asherman syndrome, 50 µg/kg/day Filgrastim was injected subcutaneously into rats at the same time of day for 5 days [29]. Then, after waiting 15 days, a hysterectomy was performed by entering the abdomen under anesthesia. For *Group 4 *(*Combined treatment group*), a total of 14 days after the establishment of the Asherman syndrome model, a second laparotomy was performed. A total of 0.01 mL hyaluronic acid was applied to the right uterine horn and, simultaneously, a subcutaneous injection of 50 μg/kg/day Filgrastim was given to the rats for 5 days (50 µg/kg/day mean for every kilogram of body weight and for every gram of tissue, there is 50 µg of a substance).

After 15 days, a hysterectomy was performed under anesthesia. All rats were sacrificed using high-dose isoflurane concentration: The rats were euthanized using 4–5% isoflurane in a closed chamber until they lost consciousness, and death was further confirmed based on cessation of breathing, ensuring minimal stress and pain. Isoflurane was administered for 3–5 min, depending on the animal’s condition, to ensure complete euthanasia. All tissues obtained from the hysterectomy were fixed in 10% formaldehyde and then used for immunohistochemical (IHC)-pathological examination.

*Pathological evaluation.* For hematoxylin and eosin staining, the shape of the endometrial epithelium, gland structure and number, uterine wall diameter, uterine cavity lumen, and inflammation level (using a semi-quantitative inflammation scoring based hematoxylin and eosin staining) were evaluated. Tissue samples from the uterine horns of each animal were fixed in 10% formalin, embedded in paraffin, and cut into 4 μm thick sections. Hematoxylin and eosin staining was performed on three representative sections per animal to evaluate histological changes. For the evaluation of histological changes, three representative sections per animal were selected systematically, taking one section out of every ten consecutive serial sections from the entire uterine tissue sample.

The morphology of endometrial glands (shape and integrity of the glandular epithelium) was qualitatively examined on H&E-stained sections, and gland counts were performed manually. Three representative sections per animal were evaluated, and the gland counts from these sections were averaged to obtain a representative gland count for each animal. Evaluations were performed under a light microscope (typically at 100× magnification).

Uterine wall thickness was measured on H&E-stained sections from the luminal epithelial surface (endometrial lining) to the outer surface of the uterus (myometrium/serosa). Measurements were consistently taken from the same anatomical region of each section. For each animal, multiple measurements (approximately three per section) were recorded on three representative sections, and their averages were calculated. Measurements were conducted in micrometers using a calibrated ocular micrometer or digital image analysis software, ensuring accuracy and objectivity.

The uterine cavity lumen diameter was defined as the widest internal lumen distance within each cross-section. This diameter was determined by measuring the maximal distance between opposite endometrial surfaces. Measurements were taken from three representative sections per animal and averaged. Quantitative morphometric measurements of uterine wall thickness and lumen diameter enabled comparison between the treatment (right horn) and control (left horn) sides.

Inflammatory cell infiltration in the endometrium and underlying stroma was evaluated semi-quantitatively on H&E-stained sections using a previously established 0–3 scoring scale for uterine tissue inflammation [28]. Scoring criteria were as follows: Score 0: No inflammation (absence of inflammatory cells). Score 1: Mild inflammation (few inflammatory cells, e.g., occasional lymphocytes or plasma cells). Score 2: Moderate inflammation (clearly present inflammatory cells; plasma cells, neutrophils, and/or eosinophils infiltrating the tissue). Score 3: Severe inflammation (dense and widespread infiltration of numerous inflammatory cells with microabscess formation).

For Masson’s trichrome staining: Fibrotic areas were evaluated using Masson’s trichrome staining by measuring the ratio of fibrotic tissue area to total uterine tissue area [(fibrotic area/analyzed uterine area) × 100]. Measurements were performed using ImageJ software (NIH, Bethesda, MD, USA, Version 1.53c). The Fibrotic area was determined as a percentage and graded as follows: Grade 0, no fibrosis; Grade 1, minimal fibrous with an increase in tissue; Grade 2, increase in irregular fibrous tissue; Grade 3, concentric fibrosis, hyalinization was used. Masson Trichrome staining method, which is a widely accepted technique for highlighting collagen fibers and visualizing fibrotic tissue. The fibrosis findings are clearly demonstrated in Figure 1B. This method has previously been described and validated in similar studies [28].

IHC staining was performed by ThermoScientific Immunostaining (Waltham, MA, USA) for anti-VEGF (cat. no. PB9071; Boster Biological Technology, Wuhan, China). The degree of staining for VEGF was scored based on positively stained glandular or stromal cells in the selected area as follows: Score 0, no staining in stromal or glandular cells; Score 1, <20% staining; Score 2, 20–60% staining; Score 3, >60%. Appropriate positive (rat ovarian tissue known to express VEGF) and negative (sections incubated without the primary antibody) controls were included for VEGF immunohistochemical staining to ensure specificity and reliability of staining results. VEGF-positive areas were identified by selecting regions exhibiting clear immunoreactivity within endometrial tissue sections. Quantitative analysis of these regions was performed using ImageJ software (NIH, Bethesda, MD, USA, Version 1.53c). VEGF expression was evaluated based on a semi-quantitative scoring system, previously validated and described in similar experimental studies [28].

All histopathological assessments were performed by two independent pathologists who were blinded to the treatment groups. In cases of discrepancy in scoring or measurements, the relevant sections were re-examined and discussed to reach a consensus. This approach ensured objectivity and reliability in the evaluation of gland counts, uterine wall thickness, lumen diameter, and inflammation scoring.

*Statistical analysis*. SPSS (IBM Corp IBM Corp., Armonk, New York, USA, Version 24.0.) was used for analysis. A 95% confidence interval was used for analysis. Descriptive statistics are presented as the mean ± the standard deviation, or the median with minimum and maximum values. A Shapiro–Wilk test was used to test for normality. A Mann–Whitney U test and the Kruskal–Wallis test analyzed pairwise group comparisons that did not show a normal distribution. *p* < 0.05 was considered to indicate a statistically significant difference.

## 3. Results

Establishment of the Asherman syndrome model was confirmed histologically following trichloroacetic acid use. As shown in Figure 1A, Asherman syndrome was established in the right horn based on the decreased number of glands, uterine wall diameter, and lumen diameter (magnification, ×100; hematoxylin and eosin). In Group 1, the establishment of the Asherman syndrome model was confirmed in two rats. Figure 1B showed the increase in fibrosis in the right horn (magnification, ×100; Masson Trichrome). The figures focus on the microscopic evaluation of the right uterine horns.

Table 1 shows the uterine wall thickness of the right and left horns. The uterine wall thickness of the right uterine horn was significantly reduced in Group 1 (*p* = 0.004), Group 2 (*p* = 0.035), and Group 4 (*p* = 0.037). Although Group 3 had a similar uterine wall thickness between the right and left horns, the difference was not significant.

Table legends:Group 1: Asherman model only (no treatment).Group 2: Received hyaluronic acid treatment.Group 3: Received filgrastim treatment.Group 4: Received both hyaluronic acid and filgrastim treatments.

The comparison of the lumen diameter in Groups 1–4 is shown in Figure 2. Table 2 shows the comparison of the right and left uterine cavities. Group 1 had a significantly decreased lumen diameter in the right uterine horn compared to the left one (*p* < 0.04). The uterine lumen diameters between the right and left horns did not differ significantly in Groups 2–4 (*p* > 0.05).

Table 3 shows the glandular counts of the right and left uterine cavities. Group 1 (*p* < 0.028) and Group 2 (*p* < 0.039) had a decreased glandular count in the right horn compared to the left. Groups 3 and 4 had similar glandular counts between the right and left uterine cavities within the respective group (*p* < 0.077 and 0.227, respectively).

Table 4 shows the results of the analysis of inflammation and fibrosis. The right uterine cavity of Groups 1 and 2 exhibited increased inflammation compared to the left uterine. Groups 3 and 4 had similar levels of inflammation between the right and left uterine cavity. Groups 1 and 2 had significantly increased fibrosis in the right uterine cavity compared to the left uterine within the respective group (*p* < 0.007 and 0.024, respectively). Groups 3 and 4 had a similar degree of fibrosis between the right and left uterine cavities, within the respective group.

Table 5 shows the results of immunohistochemical analysis of VEGF. Group 1 had significantly decreased VEGF levels in the right uterine cavity (*p* < 0.007). Groups 2–4 had similar levels of VEGF expression in the right and left uterine cavities.

## 4. Discussion

Intrauterine adhesions are a significant issue with increasing rates of cesarean sections and endometrial surgical procedures attributed to this condition, necessitating the exploration of novel treatments. Although hysteroscopic adhesiolysis is currently the preferred method, recurrence remains a notable concern. This study investigated the use of hyaluronic acid and filgrastim as a therapeutic approach in a rat model of Asherman syndrome. While hyaluronic acid treatment led to an increase only in the uterine lumen diameter and VEGF levels, Filgrastim treatment resulted in increased uterine wall thickness, lumen diameter, gland count, and VEGF levels, and decreased fibrosis and inflammation scores.

Although hysteroscopic adhesiolysis is the standard treatment for Asherman syndrome, the risk of postoperative adhesion recurrence remains a significant concern. To prevent this, estrogen therapy, intrauterine devices, foley, and intrauterine gel applications are frequently used. However, there is no consensus treatment method. Lin et al. [30] compared the effectiveness of intrauterine balloons, intrauterine contraceptive devices, and hyaluronic acid gel after hysteroscopic adhesiolysis. Among 107 women, 20 patients received intrauterine balloons, 28 received intrauterine devices, 18 received hyaluronic acid gel, and 41 received no treatment. In control hysteroscopies, adhesions significantly decreased in the intrauterine balloon and IUD groups, while the hyaluronic acid gel group was similar to the control group. In the present study, hyaluronic acid and filgrastim treatment were compared. Hyaluronic acid treatment increased uterine lumen diameter and neovascularization. However, there was no significant effect on gland count, inflammation level, fibrosis, or uterine wall diameter. Moreover, the increase in lumen diameter observed with hyaluronic acid treatment was less pronounced than that seen with Filgrastim treatment.

A meta-analysis by Unanyan et al. [31] evaluated the effects of IUD and hyaluronic acid in their meta-analysis. This research analyzed a total of 1226 participants across eight randomized controlled trials. In five different studies, hyaluronic acid gel significantly reduced adhesion formation after surgical intervention. This meta-analysis suggested that hyaluronic acid gel increased pregnancy rates, although more prospective studies are required to confirm this observation. Hooker et al. [32] conducted a multicenter prospective randomized study to investigate whether hyaluronic acid gel application after dilation and curettage reduced adhesion formation. A total of 152 women were randomly assigned to receive gel application or no additional treatment after D&C. Although the use of hyaluronic acid gel seemed to reduce IUA incidence and severity, it did not eliminate adhesion formation. In the present study, hyaluronic acid treatment increased lumen diameter and vascularization, but did not significantly increase uterine wall diameter, gland count, inflammation, or fibrosis scoring.

It has been indicated that G-CSF increases VEGF release from neutrophils, increasing angiogenesis. Additionally, the positive effect of angiogenesis on implantation has also been suggested. Gleicher et al. [19] used G-CSF as a treatment for a thin endometrium. G-CSF application significantly increased endometrial thickness but resulted in low clinical pregnancy rates (19.1%, 18.9%) [29]. G-CSF plays a role in mobilizing hematopoietic stem cells and progenitor cells [33]. G-CSF also plays a role in stimulating endometrial stem cells and promotes endometrial development through the mobilization of bone marrow stem cells [34,35]. In the present study, it was demonstrated that filgrastim treatment increased endometrial proliferation, uterine lumen diameter, uterine wall diameter, gland count, and VEGF levels, and reduced fibrosis and inflammation. Combination therapy with hyaluronic acid and Filgrastim increased lumen diameter, gland count, vascularization, and decreased fibrosis and inflammation scores, but with no effect on uterine wall thickness. Of note, the combined treatment had no additional advantage compared to the Filgrastim treatment.

Our results indicated that adding hyaluronic acid to filgrastim did not further enhance endometrial repair compared to filgrastim alone; key histological parameters such as uterine wall thickness, lumen diameter, and gland count, as well as fibrosis and inflammation scores, were statistically similar between the filgrastim-only group and the combined treatment group. This suggests that filgrastim monotherapy may have already elicited a near-maximal regenerative response, bringing these parameters close to normal levels—a ceiling effect that left little room for additional improvement by hyaluronic acid. Filgrastim exerts its effects through robust mobilization of bone-marrow–derived cells and enhanced VEGF expression, which together promote angiogenesis and tissue regeneration. In this context, hyaluronic acid, which primarily functions as a physical barrier and supportive matrix, does not provide significant additive benefit beyond the potent regenerative effects achieved by filgrastim alone.

The present study has some limitations. The sample size of the experimental animal groups was small due to undesirable losses. Increased toxicity and associated mortality were observed with the use of high doses of trichloroacetic acid, leading to higher death rates compared to other models. While chemical-agent-based models are effective in inducing endometrial damage, they can result in severe tissue injury and systemic complications in rats. Another limitation was that the present study did not assess whether Filgrastim affected pregnancy outcomes. As another limitation of our study, histological images of fibrosis formation were not collected in groups other than the Asherman model validation group. This is due to the necessity of preserving the uterine horns for subsequent analyses in order to maintain the integrity of the experimental design. Additionally, the findings from animal experiments may not directly translate to humans. The effects of the given treatment modalities may vary in different species. Lastly, the low resolution of the microscope used in this study is acknowledged as a limitation, which may have affected the clarity and detail of the histological images, and it is not possible to provide additional examples from each group, diagrams, or further details about stained tissues due to the limitations of our data and resources.

In conclusion, re-adhesion formation after adhesiolysis is an important problem in Asherman syndrome. Novel treatment modalities that are feasible and cost-effective to prevent fibrosis are required. In the present study, uterine cavities in a rat model of Asherman syndrome were compared with healthy uterine cavities. Hyaluronic acid treatment promoted a lumen diameter and vascularity close to that of the healthy uterine cavity, but the number of glands remained lower and inflammation and fibrosis remained higher than the healthy controls. Filgrastim treatment resulted in lumen diameter, uterine wall thickness, vascularity, number of glands, inflammation, and fibrosis scores similar to the healthy controls. The results of this study suggested that Filgrastim may serve as a novel treatment modality for the management of Asherman syndrome, although these results need to be supported by clinical studies.

## Figures and Tables

**Figure 1 jcm-14-03334-f001:**
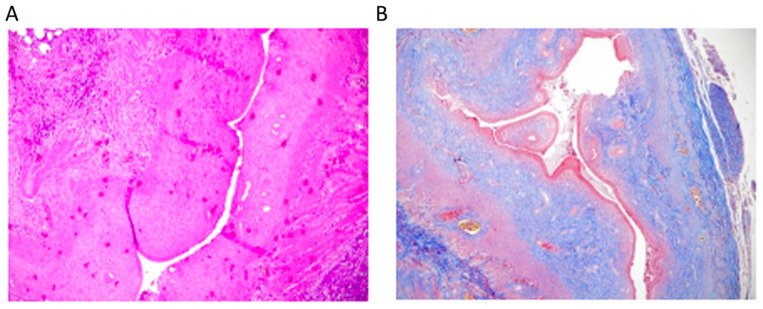
(**A**) Model of Asherman syndrome in the right horn. Analysis showed a decreased number of glands, a decreased uterine wall diameter, and a smaller lumen diameter. Hematoxylin and eosin stain. Magnification, ×100. (**B**) In Group 1, the Asherman syndrome model in the right horn exhibited increased fibrosis as evidenced by Masson Trichrome staining (Magnification, ×100), serving solely to confirm the establishment of Asherman syndrome in the two rats. Scale bar = 100 µm.

**Figure 2 jcm-14-03334-f002:**
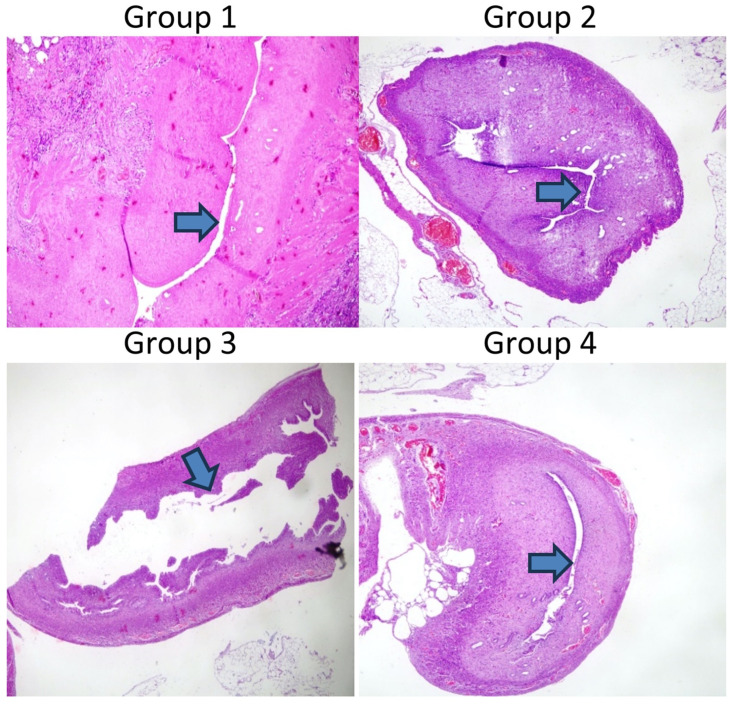
Hematoxylin and eosin stain of right horn. Comparison of the lumen diameter in the different groups. Magnification, ×100. The arrows in Figure 2 indicate the lumen diameter, highlighting the differences observed between groups. Scale bar = 100 µm.

**Table 1 jcm-14-03334-t001:** Comparison of uterine wall thickness of the right and left uterine horns.

Group	Mean ± SD	Median (Min–Max)	*p*-Value
Group 1			0.004
Right	892.5 ± 668.2	892.5 (420–1365)	
Left	1879.5 ± 311.8	1879.5 (1659–2100)	
Group 2			0.035
Right	1041.4 ± 560.4	980 (430–1800)	
Left	1915.3 ± 767	1870 (1100–3100)	
Group 3			0.078
Right	1356.2 ± 623.5	1190 (798–2150)	
Left	2151.5 ± 893.1	2124.5 (850–3488)	
Group 4			0.037
Right	948.5 ± 350.2	788.5 (670–1515)	
Left	1652.5 ± 603.3	1550 (1085–2670)	

**Table 2 jcm-14-03334-t002:** Comparison of the right and left uterine cavity lumen diameter.

Group	Mean ± SD	Median (Min–Max)	*p*-Value
Group 1			0.004
Right	198.5 ± 16.3	198.5 (187–210)	
Left	810 ± 381.8	810 (540–1080)	
Group 2			0.064
Right	500.4 ± 350.7	410 (188–1210)	
Left	908.6 ± 373.3	980 (370–1490)	
Group 3			0.150
Right	727.5 ± 408.2	601.5 (286–1280)	
Left	1153.7 ± 573.2	1167.5 (482–1960)	
Group 4			0.150
Right	462.8 ± 269.8	410.5 (205–906)	
Left	714.8 ± 353.6	665.5 (379–1189)	

**Table 3 jcm-14-03334-t003:** Comparison of uterine glandular counts in the right and left uterine cavities.

Group	Mean ± SD	Median (Min–Max)	*p*-Value
Group 1			0.029
Right	7.0 ± 2.8	7.0 (5–9)	
Left	12.5 ± 5.0	12.5 (9–16)	
Group 2			0.039
Right	6.3 ± 2.2	6.0 (3–9)	
Left	10.1 ± 3.9	9.0 (7–18)	
Group 3			0.077
Right	9.3 ± 3.1	8.5 (6–14)	
Left	13.2 ± 3.4	12.5 (9–18)	
Group 4			0.227
Right	8.0 ± 4.0	7.5 (4–13)	
Left	11.2 ± 4.7	10.0 (6–19)	

**Table 4 jcm-14-03334-t004:** Comparison of the inflammation and fibrosis levels between the right and left uterine cavity.

Group	Inflammation Levels			Fibrosis		
	Mean ± SD	Median (Min–Max)	*p*-Value	Mean ± SD	Median (Min–Max)	*p*-Value
Group 1			0.050			0.007
Right	2.5 ± 0.7	2.5 (2–3)		2.5 ± 0.7	2.5 (2–3)	
Left	1.5 ± 0.7	1.5 (1–2)		1.5 ± 0.7	1.5 (1–2)	
Group 2			0.019			0.024
Right	1.6 ± 0.8	1 (1–3)		2.1 ± 0.7	2 (1–3)	
Left	0.6 ± 0.5	1 (0–1)		1.1 ± 0.7	1 (0–2)	
Group 3			0.203			0.485
Right	1.3 ± 1.2	1.5 (0–3)		1.2 ± 1.2	1 (0–3)	
Left	0.5 ± 0.6	0.5 (0–1)		0.7 ± 0.5	1 (0–1)	
Group 4			0.116			0.051
Right	1.5 ± 0.6	1.5 (1–2)		1.8 ± 0.8	2 (1–3)	
Left	0.8 ± 0.8	1 (0–2)		0.8 ± 0.8	1 (0–2)	

**Table 5 jcm-14-03334-t005:** Comparison of Vascular Endothelial Growth Factor staining in the right and left uterine cavity.

Group	Mean ± SD	Median (Min–Max)	*p*-Value
Group 1			0.011
Right	1.5 ± 0.7	1.5 (1–2)	
Left	2.5 ± 0.7	2.5 (2–3)	
Group 2			0.207
Right	1.9 ± 0.7	2 (1–3)	
Left	1.4 ± 0.8	1 (1–3)	
Group 3			0.268
Right	1.8 ± 0.8	2 (1–3)	
Left	2.3 ± 0.8	2.5 (1–3)	
Group 4			0.176
Right	1.7 ± 0.8	1.5 (1–3)	
Left	2.3 ± 0.8	2.5 (1–3)	

## Data Availability

The datasets generated and analyzed during the current study are available from the corresponding author upon reasonable request.

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
