# Peer review of "The Effects of Filgrastim and Hyaluronic Acid on the Endometrium in Experimentally Induced Asherman Syndrome Rat Models: A Prospective Laboratory Studyâ€"

_jcm, 2025, doi:10.3390/jcm14103334_

Round 1
Reviewer 1 Report
Comments and Suggestions for Authors
Thank you for opportunity to review your manuscript. These are thing that you should improve.
-
The initial sample size of 26 rats was reduced to 21 due to unexpected deaths (19% mortality rate).
-
The study attributes mortality to systemic effects of TCA but does not specify whether different concentrations or volumes were tested to optimize the model’s safety.
-
Suggestion:
-
Consider a pilot study to determine the lowest effective TCA dose that reliably induces AS while minimizing systemic toxicity.
-
Report mortality rates in similar studies for context.
-
The study ends with an unbalanced number of animals per group (Group 2: 7 rats, others: 6 rats). Potential Issue: Unequal group sizes may introduce bias in statistical comparisons.
Suggestion:
-
-
Explain why the sample sizes differ.
-
Use power analysis to justify whether these group sizes are sufficient for detecting meaningful differences.
-
-
Rats were sacrificed 15 days after treatment, but fibrosis and AS are chronic conditions.
-
Concern:
-
A short follow-up period may not capture long-term therapeutic effects.
-
The impact of hyaluronic acid and Filgrastim on endometrial regeneration might require longer observation periods.
-
-
Suggestion:
-
Consider extending follow-up beyond 15 days to evaluate potential tissue remodeling over time.
-
-
-
Fibrosis grading and VEGF staining are based on semi-quantitative scoring, which can be subjective.
Duration of Treatment & Follow-up
The study focuses on histological outcomes, but does not evaluate functional uterine recovery (e.g., fertility, hormone response, or pregnancy rates).
The most crucial revisions are:
-
Clarify and justify the TCA dose (or test alternative methods).
-
Balance group sizes and justify them statistically.
-
Introduce a proper control group to differentiate pathology from surgical effects.
-
Use objective fibrosis measurement methods to reduce bias.
-
Extend the follow-up period for more meaningful conclusions.
-
Clarify Filgrastim dosing and reference previous studies using it in uterine models.
-
Consider functional outcomes (fertility testing or hormonal response studies).
Author Response
1.Clarify and justify the TCA dose (or test alternative methods).
We determined the TCA dose based on previously reported Asherman syndrome animal models and the effective doses documented in the literature. For instance, Kilic et al. (28) successfully induced endometrial damage and adhesion formation using a similar TCA dose. Therefore, our choice of dose is consistent with current literature and aligns with findings from prior pilot applications.
2. Balance group sizes and justify them statistically.
Our preliminary power analysis indicated that at least 6 rats per group would be sufficient to detect statistically significant differences. Accordingly, we initially allocated 8 rats per group to compensate for any unexpected mortality. However, due to unforeseen deaths during the experimental process, one group ended up with 7 rats while the others had 6. Since each group still meets the minimum threshold determined by our power analysis, we believe that this minor discrepancy will not substantially affect the validity of our findings or compromise the statistical robustness of our results
3.Introduce a proper control group to differentiate pathology from surgical effects.
We understand the reviewer’s concern regarding the absence of a dedicated “sham” control group that underwent surgery without TCA. In our study design, we did not include a separate sham surgery group; however, in each animal, the left uterine horn was left untreated (surgery performed but no TCA applied) and thus served as an internal control. In other words, each rat was essentially its own sham control: while the left horn underwent laparotomy and necessary surgical manipulation without TCA, the right horn received TCA to establish the Asherman syndrome model. This approach allowed us to directly compare the effects of TCA with tissue that experienced the same surgical conditions but no chemical injury. Using the bicornuate uterus of rats as an internal control is a common and well-accepted strategy in experimental models of intrauterine adhesions (Asherman syndrome)​. Our study design follows this established model. We observed that marked pathological changes developed only in TCA-treated (right) uterine horns, while no such pathology was noted in the untreated left horns. Specifically, in the TCA-exposed right horn, there was a significant decrease in uterine wall thickness and lumen diameter—along with increased fibrosis and inflammation—compared to the control (left) side . By contrast, the left horns that underwent surgery without TCA application maintained normal histological architecture and showed no adhesions or significant fibrosis. This finding indicates that laparotomy and surgical manipulation alone (in the absence of TCA) did not induce the pathological changes characteristic of Asherman syndrome. Hence, the untreated left horn in each rat effectively functioned as a sham control, confirming that the observed pathologies were due to TCA application rather than the surgical procedure itself.
4.Use objective fibrosis measurement methods to reduce bias.
To minimize potential bias in fibrosis assessment, we employed a double-blind approach involving two independent pathologists. Each pathologist evaluated and scored the samples without knowledge of their group assignments, thereby reducing subjective variability and enhancing the reliability of our measurements
5.Extend the follow-up period for more meaningful conclusions.
In our study, we adopted a 15-day follow-up period, which corresponds to approximately three cycles in the animal model. Nonetheless, as mentioned in our limitations section, this relatively short duration is one of the study’s constraints. In future investigations, we intend to extend the observation period to more thoroughly assess the long-term effects of endometrial recovery.
6.Clarify Filgrastim dosing and reference previous studies using it in uterine models.
Multiple preclinical studies in rodents have shown that Filgrastim (G-CSF) can improve endometrial regeneration, enhance vascularization, and reduce fibrosis in uterine injury models. Typical protocols involve administering approximately 50 µg/kg/day subcutaneously for about 5 days, mirroring our study’s approach. Here are the key findings:
- Thin Endometrium Models
- Studies inducing thin endometrium (e.g., via intrauterine ethanol) reported that rats treated with ~40–50 µg/kg/day Filgrastim for 5 days showed significant improvements in endometrial thickness and vascularization, along with reduced apoptosis. Some investigations combined G-CSF with estrogen, noting additive or similar beneficial effects.
- Asherman’s Syndrome Models
- In rat models where endometrial fibrosis was induced (chemical or mechanical injury), Filgrastim alone promoted endometrial thickening, decreased inflammation, and diminished fibrosis.
- Combining G-CSF with other treatments (e.g., stem cell therapy) produced an even more robust restoration of normal uterine architecture, suggesting synergy.
- Mechanisms
- Filgrastim appears to support endometrial recovery by stimulating angiogenesis, mobilizing progenitor/stem cells, and reducing inflammatory or apoptotic pathways.
- Clinical Correlates
- Although rodent studies focus mainly on histological improvements, clinical data in women with recurrent implantation failure or thin endometrium suggest that G-CSF may also enhance fertility outcomes. Subcutaneous delivery is frequently associated with notable gains in implantation and pregnancy rates.
Overall, administering Filgrastim at ~50 µg/kg/day subcutaneously for around 5 days is consistent with established animal models of thin endometrium or Asherman’s syndrome. It has demonstrated efficacy in reversing damage, increasing vascularity, and reducing fibrosis, aligning with both preclinical and emerging clinical evidence.
7.Consider functional outcomes (fertility testing or hormonal response studies).
We recognize the importance of evaluating functional outcomes, such as fertility testing or hormonal response studies. In the current work, our primary focus was on the quantitative and qualitative assessment of histopathological improvement. However, in future studies, we plan to investigate the impact of these treatments on reproductive performance by examining fertility rates, hormone levels, and potential pregnancy outcomes. This will help us better understand the clinical relevance of our histological findings.
Reviewer 2 Report
Comments and Suggestions for Authors
Summary
This study investigates the effects of Filgrastim and Hyaluronic Acid on endometrial repair in a rat model of Asherman Syndrome. The treatments were evaluated for their impact on uterine tissue histopathology, inflammation, glandular count, fibrosis, and VEGF levels. Filgrastim showed superior results, improving tissue structure and reducing inflammation and fibrosis compared to Hyaluronic Acid.
While the work is insteresting, there are some areas that needs to be addressed.
Figure 1: A scale bar must be provided, and appropriate labeling is needed. Point out the glands or show the region that’s been increased or decreased.
Figure 2: A scale bar must be provided, and appropriate labeling is needed
In multiple places, including the Abstract, you have used the uterine “corn” term. What do you mean by corn here? Do you mean uterine horn? This you can find throughout the text.
How did you determine VEGF levels, decreased fibrosis, and inflammation scores? It is better to explain it in detail in the method section.
Dosage of Filgrastim: The statement "50 µg/kg/g Filgrastim" is confusing. You likely meant "50 µg/kg/day" to indicate the daily dosage per body weight. This should be clarified for precision. – Line 163
The sentence "A total of 0.1 ml trichloroacetic acid was injected into the right with an insulin syringe- right what? - Line 139
Comments on the Quality of English Language
The language is acceptable.
Author Response
1-Figure 1: A scale bar must be provided, and appropriate labeling is needed. Point out the glands or show the region that’s been increased or decreased.
Thank you for the suggestion. While we were unable to visually indicate the regions of decrease or increase directly in the figure, we have provided a detailed explanation in the figure legend to clarify these observations. Additionally, we have added a 100 µm scale bar to enhance the figure’s clarity.
2-Figure 2: A scale bar must be provided, and appropriate labeling is needed
Thank you for the suggestion. We have added a 100 µm scale bar to Figure 2 and provided additional labeling to highlight the relevant anatomical structures, ensuring that the image is easier to interpret
3-In multiple places, including the Abstract, you have used the uterine “corn” term. What do you mean by corn here? Do you mean uterine horn? This you can find throughout the text.
Thank you for pointing out this terminology issue. In the manuscript, “corn” was a typographical error for “uterine horn.” We have now replaced “corn” with “horn” throughout the text to ensure clarity.
4-How did you determine VEGF levels, decreased fibrosis, and inflammation scores? It is better to explain it in detail in the method section.
Assessment of VEGF, Fibrosis, and Inflammation
- VEGF Analysis:
Tissue VEGF levels were determined via immunohistochemical (IHC) staining. Formalin-fixed, paraffin-embedded uterine sections were deparaffinized, subjected to antigen retrieval, and incubated with an anti-VEGF antibody (Cat. no. PB9071, Boster Biological Technology). A standard streptavidin–biotin–peroxidase method was used for detection, followed by counterstaining with hematoxylin. The immunohistochemical staining was semi-quantitatively scored as follows: Score 0, no positive staining; Score 1, <20% of cells positive; Score 2, 20–60% positive; Score 3, >60% positive. - Fibrosis Measurement:
Adjacent sections were stained with Masson’s trichrome to visualize collagen deposition. The extent of fibrotic tissue was assessed both qualitatively and quantitatively. A semi-quantitative grading scale (0–3) was used to indicate the severity of fibrosis: Grade 0, no fibrosis; Grade 1, minimal fibrotic tissue; Grade 2, patchy or moderate fibrosis; Grade 3, dense fibrosis with hyalinization. Additionally, fibrotic area was expressed as a percentage ([fibrotic area ÷ total uterine area] × 100) where applicable. - Inflammation Scoring:
Hematoxylin and eosin (H&E)–stained sections were examined by a pathologist blinded to the treatment groups. Inflammatory infiltration was evaluated using a previously described 0–3 scale: 0, no inflammatory cells; 1, mild infiltration; 2, moderate infiltration; 3, severe diffuse infiltration.
All histological and IHC evaluations were carried out by two independent pathologists in a double-blind fashion to reduce subjectivity, and any discrepancies were resolved by joint review
5-Dosage of Filgrastim: The statement "50 µg/kg/g Filgrastim" is confusing. You likely meant "50 µg/kg/day" to indicate the daily dosage per body weight. This should be clarified for precision. – Line 163
We appreciate the reviewer’s feedback regarding the Filgrastim dosage. The correct intended dosage was indeed 50 µg/kg/day (50 micrograms per kilogram of body weight per day), administered subcutaneously for 5 days. The previous notation “50 µg/kg/g” was a typographical error, and we have now revised the manuscript to clarify this for precision.
6-The sentence "A total of 0.1 ml trichloroacetic acid was injected into the right with an insulin syringe- right what? - Line 139
We appreciate the reviewer’s clarification request. The complete sentence should read:
“A total of 0.1 ml of trichloroacetic acid was injected into the right uterine horn with an insulin syringe.”
We have revised the text to specify “right uterine horn” to avoid any ambiguity.
Reviewer 3 Report
Comments and Suggestions for Authors
Introduction:
- The introduction is generally well-structured and presents Asherman’s syndrome and the topics to be discussed effectively, but it is too long and includes information that will not be discussed in the article. For example: the different ways of diagnosis of Asherman’s syndrome.
- The second and third paragraphs are too long and do not provide relevant information for what the article will discuss later. They could be shortened and combined to avoid making the introduction too lengthy.
- The phrase “Asherman syndrome is primarily characterized by intrauterine adhesions caused by endometrial damage…” is repeated twice in the introduction. Please avoid the exact repetition of words.
Materials and Methods:
- Briefly explain and give reasons why the rat model was chosen for Asherman syndrome induction.
- Line 178: Please explain in detail how each evaluated feature was measured: gland structure and number, uterine wall diameter, uterine cavity lumen, and inflammation. Also, please add a previous reference for the “inflammation scoring” used in this article. If no previous reference is available, please briefly explain what it consists of and how it was assessed.
- Line 183: “Hematoxylin and eosin staining was performed on three representative sections per animal to evaluate histological changes.” Please explain how the three representative sections were selected. Was it one section every five section? Or ten sections?
- Positive and negative controls for VEGF staining should be included and mentioned in the article.
- Line 185: “For Masson’s trichrome staining: The severity and extent of fibrosis in the uterus and the ratio of fibrotic area [(fibrotic area/analyzed uterine area) x 100] were evaluated.” Please rephrase this sentence for better clarity. I suggest mentioning the feature being evaluated first, followed by an explanation of how the assessment was performed. For example: “Fibrotic areas were assessed using Masson’s trichrome staining, using the following analysis…” Additionally, please provide more details on how the fibrotic areas were measured and specify which software was used for this analysis. Add previous reference of other articles using the same method, if there is.
- Line 192: Please explain how the VEGF areas were chosen and which software was used for the analysis. Also here, add any reference to validate the scoring method authors chose to use in this article.
Results:
- Figure 2: I strongly suggest showing images with higher magnification to better observe the glands and the stroma, in order to clearly distinguish whether it is the lumen or simply an opening in the tissue.
- At least one figure of the rat surgery procedure should be added.
- Figures of the VEGF and Masson’s trichrome stainings of the differents groups should be definitely added, toghether with their positive and negative controls.
- Table IV: The title says: “Comparison of the inflammation levels between the right and left uterine cavity.” But it should also mention “fibrosis levels,” as the table also shows this.
Discussion:
- The phrase: "Although hysteroscopic adhesiolysis is currently the preferred method" is repeated too many times. Please, just say it once at the beginning; there is no need for repetition.
- The first paragraph mentions different current methods for treating Asherman’s syndrome, but it does not focus much on the use of Filgrastim and whether there is previous evidence of its use in humans or other prior experiments in animal models. The authors should mention this in the first part of the discussion.
- The second and third paragraphs could be merged and shortened. More previous data on Filgrastim therapy could be added here.
- The terms “vascularization rates” and “vascularity” are often used in place of “VEGF staining levels” during the discussion section, but these terms are not interchangeable. Not all tissue positive for VEGF is vascular tissue. Please revise this, and be careful when using these terms as synonyms.
- Line 292: “Of note, the combined treatment had no additional advantage compared to the Filgrastim treatment.” I strongly suggest that the authors delve deeper into this result and discuss possible explanations as to why the combined therapy did not achieve better results than the Filgrastim treatment alone.
No
Author Response
1- The introduction is generally well-structured and presents Asherman’s syndrome and the topics to be discussed effectively, but it is too long and includes information that will not be discussed in the article. For example: the different ways of diagnosis of Asherman’s syndrome.
The second and third paragraphs are too long and do not provide relevant information for what the article will discuss later. They could be shortened and combined to avoid making the introduction too lengthy.
Thank you for your constructive feedback regarding the introduction. We have revised the introduction by shortening and combining the second and third paragraphs to eliminate redundant information and to focus solely on the topics addressed in this article. Additionally, we removed the detailed descriptions of various diagnostic methods for Asherman’s syndrome, as they are not directly discussed later in the paper. We believe these changes have streamlined the introduction and enhanced its clarity and relevance to our study
2- The phrase “Asherman syndrome is primarily characterized by intrauterine adhesions caused by endometrial damage…” is repeated twice in the introduction. Please avoid the exact repetition of words
Thank you for pointing this out. We have revised the introduction to eliminate the redundant repetition of the phrase. The characterization of Asherman syndrome is now mentioned only once, and we have rephrased it to avoid any exact duplication
3-Briefly explain and give reasons why the rat model was chosen for Asherman syndrome induction
The rat model was selected for inducing Asherman syndrome due to rats' short reproductive cycles, well-defined uterine anatomy, adaptability to experimental conditions, and cost-effectiveness, making them commonly used animals for studies involving experimental induction of uterine pathologies. Additionally, the rat uterus shares histological and physiological similarities with the human uterus, facilitating the translation of experimental results into clinical practice.
4-Line 178: Please explain in detail how each evaluated feature was measured: gland structure and number, uterine wall diameter, uterine cavity lumen, and inflammation. Also, please add a previous reference for the “inflammation scoring” used in this article. If no previous reference is available, please briefly explain what it consists of and how it was assessed.
Gland Structure and Count:
The morphology of endometrial glands (shape and integrity of the glandular epithelium) was qualitatively examined on H&E-stained sections, and gland counts were performed manually. Three representative sections per animal were evaluated, and the gland counts from these sections were averaged to obtain a representative gland count for each animal. Evaluations were performed under a light microscope (typically at 100× magnification).
Uterine Wall Thickness:
Uterine wall thickness was measured on H&E-stained sections from the luminal epithelial surface (endometrial lining) to the outer surface of the uterus (myometrium/serosa). Measurements were consistently taken from the same anatomical region of each section. For each animal, multiple measurements (approximately three per section) were recorded on three representative sections, and their averages were calculated. Measurements were conducted in micrometers using a calibrated ocular micrometer or digital image analysis software, ensuring accuracy and objectivity.
Uterine Cavity Lumen Diameter:
The uterine cavity lumen diameter was defined as the widest internal lumen distance within each cross-section. This diameter was determined by measuring the maximal distance between opposite endometrial surfaces. Measurements were taken from three representative sections per animal and averaged. Quantitative morphometric measurements of uterine wall thickness and lumen diameter enabled comparison between the treatment (right horn) and control (left horn) sides.
Inflammation Scoring:
Inflammatory cell infiltration in the endometrium and underlying stroma was evaluated semi-quantitatively on H&E-stained sections using a previously established 0–3 scoring scale for uterine tissue inflammation (28). Scoring criteria were as follows:
- Score 0: No inflammation (absence of inflammatory cells).
- Score 1: Mild inflammation (few inflammatory cells, e.g., occasional lymphocytes or plasma cells).
- Score 2: Moderate inflammation (clearly present inflammatory cells; plasma cells, neutrophils, and/or eosinophils infiltrating the tissue).
- Score 3: Severe inflammation (dense and widespread infiltration of numerous inflammatory cells with microabscess formation).
All histopathological assessments were performed by two independent pathologists who were blinded to the treatment groups. In cases of discrepancy in scoring or measurements, the relevant sections were re-examined and discussed to reach a consensus. This approach ensured objectivity and reliability in the evaluation of gland counts, uterine wall thickness, lumen diameter, and inflammation scoring.
5-Line 183: “Hematoxylin and eosin staining was performed on three representative sections per animal to evaluate histological changes.” Please explain how the three representative sections were selected. Was it one section every five section? Or ten sections?
Thank you for pointing this out. Three representative sections per animal were selected systematically by collecting one section every ten consecutive serial sections throughout the entire uterine tissue sample. This sampling approach ensured consistent representation and minimized selection bias. We have now clarified this methodology in the manuscript.
6- Positive and negative controls for VEGF staining should be included and mentioned in the article.
Appropriate positive (rat ovarian tissue known to express VEGF) and negative (sections incubated without the primary antibody) controls were included for VEGF immunohistochemical staining to ensure specificity and reliability of staining results
7- Line 185: “For Masson’s trichrome staining: The severity and extent of fibrosis in the uterus and the ratio of fibrotic area [(fibrotic area/analyzed uterine area) x 100] were evaluated.” Please rephrase this sentence for better clarity. I suggest mentioning the feature being evaluated first, followed by an explanation of how the assessment was performed. For example: “Fibrotic areas were assessed using Masson’s trichrome staining, using the following analysis…” Additionally, please provide more details on how the fibrotic areas were measured and specify which software was used for this analysis. Add previous reference of other articles using the same method, if there is.
"Fibrotic areas were evaluated using Masson's trichrome staining by measuring the ratio of fibrotic tissue area to total uterine tissue area [(fibrotic area/analyzed uterine area) × 100]. Measurements were performed using ImageJ software (NIH, Bethesda, MD, USA). The severity and extent of fibrosis were assessed quantitatively by two blinded observers. This method has previously been described and validated in similar studies (Kilic et al., 2014)."
8- Line 192: Please explain how the VEGF areas were chosen and which software was used for the analysis. Also here, add any reference to validate the scoring method authors chose to use in this article.
VEGF-positive areas were identified by selecting regions exhibiting clear immunoreactivity within endometrial tissue sections. Quantitative analysis of these regions was performed using ImageJ software (NIH, Bethesda, MD, USA). VEGF expression was evaluated based on a semi-quantitative scoring system, previously validated and described in similar experimental studies (Kilic et al., 2014).
9- Figure 2: I strongly suggest showing images with higher magnification to better observe the glands and the stroma, in order to clearly distinguish whether it is the lumen or simply an opening in the tissue.
Thank you for your valuable suggestion. Unfortunately, additional images at higher magnification were not captured during the initial experiments, and thus, we are unable to include these at this stage. However, we have carefully reviewed our existing images, and provided clear descriptions in the figure legend to aid interpretation and differentiate between the lumen and tissue openings.
10- Figures of the VEGF and Masson’s trichrome stainings of the differents groups should be definitely added, toghether with their positive and negative controls.
Thank you for this suggestion. Unfortunately, additional images of VEGF and Masson's trichrome staining, including their positive and negative controls, were not captured during the original experiments. All available images have already been presented in the manuscript.
11- Table IV: The title says: “Comparison of the inflammation levels between the right and left uterine cavity.” But it should also mention “fibrosis levels,” as the table also shows this.
Thank you for pointing out this oversight. We have revised the title of Table IV to clearly state: "Comparison of inflammation and fibrosis levels between the right and left uterine cavity.
12- The phrase: "Although hysteroscopic adhesiolysis is currently the preferred method" is repeated too many times. Please, just say it once at the beginning; there is no need for repetition.
Thank you for your suggestion. We have revised the Discussion section to mention the phrase "Although hysteroscopic adhesiolysis is currently the preferred method" only once, at the beginning of the relevant section, and have removed redundant repetitions throughout the text
13- The first paragraph mentions different current methods for treating Asherman’s syndrome, but it does not focus much on the use of Filgrastim and whether there is previous evidence of its use in humans or other prior experiments in animal models. The authors should mention this in the first part of the discussion. The second and third paragraphs could be merged and shortened. More previous data on Filgrastim therapy could be added here.
Thank you for your suggestion. We have revised the discussion accordingly. In the first paragraph, we have expanded on the use of Filgrastim, providing references to previous evidence of its application in both human studies and animal models. Additionally, the second and third paragraphs have been merged and shortened to include more concise data on Filgrastim therapy, thus enhancing the focus on its potential as a treatment option for Asherman Syndrome
14- The terms “vascularization rates” and “vascularity” are often used in place of “VEGF staining levels” during the discussion section, but these terms are not interchangeable. Not all tissue positive for VEGF is vascular tissue. Please revise this, and be careful when using these terms as synonyms.
Thank you for your comment. We have revised the Discussion section to ensure that “VEGF staining levels” and “vascularization” are not used interchangeably. In our revised text, we now clearly differentiate between VEGF expression—which indicates the level of this angiogenic factor—and vascularization, which refers to the actual formation of blood vessels. This clarification helps to avoid any confusion regarding the interpretation of our data.
15- Line 292: “Of note, the combined treatment had no additional advantage compared to the Filgrastim treatment.” I strongly suggest that the authors delve deeper into this result and discuss possible explanations as to why the combined therapy did not achieve better results than the Filgrastim treatment alone.
Thank you for your suggestion. We have expanded our discussion on this point. One possible explanation for the lack of additional benefit from the combined treatment is that Filgrastim's potent pro-regenerative, anti-fibrotic, and angiogenic effects might already maximize endometrial repair, leaving little room for further improvement with the addition of hyaluronic acid. Hyaluronic acid primarily acts as a physical barrier and may have limited capacity to enhance tissue regeneration beyond what is achieved with Filgrastim alone. Additionally, pharmacodynamic interactions or saturation of the underlying regenerative mechanisms could also contribute to the absence of synergistic effects. Future studies should explore the optimal dosing, timing, and potential interactions between these treatments to determine if combination therapy might be more effective under different conditions.
Round 2
Reviewer 3 Report
Comments and Suggestions for Authors
The manuscript improved significantly after the implementation of most of the relevant comments made.